# 'Nexus' Narratives in Urban Vulnerable Places: Pathways to Sustainability via Municipal Health Programs in Brazil

Alberto Matenhauer Urbinatti [1,*], Simone Ley Omori-Honda [1], Carolina Monteiro de Carvalho [2], Klaus Frey [3], Pedro Roberto Jacobi [2] and Leandro Luiz Giatti [1,4]

1. School of Public Health, University of São Paulo, São Paulo 01246-904, Brazil
2. Institute of Energy and Environment, University of São Paulo, São Paulo 05508-010, Brazil
3. Center for Engineering, Modeling and Applied Social Sciences (CECS), Federal University of ABC, São Bernardo do Campo 09210-580, Brazil
4. Instituto Leônidas e Maria Deane, Fiocruz, Manaus 69057-70, Brazil
* Correspondence: albertourbinatti@gmail.com

**Abstract:** In recent years, the water–energy–food (WEF) nexus approach has been widely used as a framework in the context of urban Sustainability. However, some elements of the approach are normative, leading to a technical view of resources and technocratic policy implementation. To avoid such tendencies, this study uses the framework of 'nexus of humility'. We used insights from the Science and Technology Studies to better assess the interactions between water, energy, and food, and consider the social construction aspects of the nexus itself. The approach of Pathways to Sustainability is combined with this framework to analyze two government programs in the cities of São Paulo and Guarulhos, Brazil; namely, the Green and Healthy Environments Program and the Environmental Health Program, respectively. We interviewed 20 individuals linked to these policies and analyzed narratives inductively and deductively. The results showed six groups of narratives, namely: *environmental and social determinants of health*, *health prevention and promotion*, *intersectorality*, *politics and economy*, *territory*, *learning*, and *participation*. Moreover, we concluded that narratives related to the WEF nexus, even if not explicitly part of the government guidelines, are present within the existing axes of action. Public health was understood as an important support pillar for the development of synergies related to Sustainability in urban areas. Finally, we sought to contribute to the literature by showing how this new framework can 'open up' avenues for sustainability within the contexts of high urban vulnerability and social inequality.

**Keywords:** environmental health policy; urban vulnerability; public health system; sustainability governance; water–energy–food nexus

## 1. Introduction

Large urban systems, such as metropolitan regions, have seen an increase in socio–environmental vulnerability [1]. Over the coming decades, urban vulnerability associated with global environmental changes will demand refocusing on health aspects, including reducing rates of diseases such as malaria and dengue fever, while improving access to health services. It is estimated that approximately one-third of the global urban population, mainly in developing countries, lives in conditions similar to those in slums, which are characterized by low-income households and deficiencies in access to basic health services [2].

The so-called social determinants of health, when associated with environmental issues, constitute an important agenda that should be incorporated by all countries, in a cross-sectoral manner, into the health and environmental sector of governments [3,4]. According to Sobral and Freitas [5], social determinants must be considered to better understand the health-disease process as a result of the way society is organized and, more than that, of the conditions in which social life is constructed.

Considering the environmental and social determinants of health, strategic political actions comprise the promotion of healthy spaces, the empowerment of citizens, the development of skills, and knowledge and attitudes supportive of public health [6].

These demands are also increasingly present in the sustainability agenda [7]. We follow the suggestion of Leach et al. [8] to differentiate 'sustainability' ('*the general capability to maintain any unspecified feature of system structure or function over indefinite periods of time*', p. 18) from Sustainability ('*the capability of maintaining specified values of human well-being, social equity and environmental quality over indefinite periods of time*', p. 18). This perspective on Sustainability, resorting in this definition to the three pillars of sustainable development, but replacing economic viability with human well-being, seems suitable to host the debate on the nexus research and its relation to health issues [9].

In the context of deprived urban peripheries, the impossibility of consciously opting for sustainable consumption, including healthy eating, and the shortcomings in water and energy supply, sheds light on the so-called 'nexus of exclusion' [10]. In general terms, the water–energy–food (WEF) nexus approach seeks to invest in sustainable ecosystem services, creating more with less, and accelerating access of low-income populations to basic water, energy, and food services [11,12]. The sustainable development and transformation agendas for cities in the Global South may not only consist of large-scale interventions to improve infrastructure, but also of more modest and cumulative approaches to understanding behavioral patterns in the 'urban nexus' of the poorest populations [13].

In this sense, the first objective of this paper is to analyze thematically two municipal health programs in the São Paulo Metropolitan Region as case studies: the Green and Healthy Environment Program (GHEP) in São Paulo, and the Environment Health Program (EHP) in the city of Guarulhos. The second objective is to verify the contributions of these programs as narratives to the debate on the concept of 'pathways to Sustainability' (PTS) and WEF nexus in the Brazilian context. We hypothesize that the intersectoral characteristics of Brazilian public health policies, by encompassing environmental issues, are opportune ways to promote what we will call the 'nexus of Sustainability' in urban peripheries. The manuscript is structured as follows: Section 2 presents the conceptual framework of the research, Section 3 the methodology, Section 4 provides the results, Section 5 discusses the findings, and Section 6 presents the final conclusions.

## 2. Conceptual Framework

Recent years have testified to the rapidly growing prominence of the notion of a 'WEF nexus' across different agendas. Unlike the narrower normative approaches to nexuses, this study aims to develop narratives from the perspective of social constructivism. More specifically, we use insights from the Science and Technology Studies. The 'nexus of humility' framework (Figure 1) has been suggested for this purpose [14] to recognize the many hybrid dimensions in contemporary nexus ideas. These dimensions are constituted by the complexity generated by each type of individual and the integrative framing of the interdependencies between water, energy, and food. This conceptual construction is based on Jasanoff's concept of 'technologies of humility' [15,16] and the 'nexus of humility' of Urbinatti et al. [14]. The last proceeded with a bibliographic search conducted through the Scopus platform on terms related to science, policy, and 'nexus', selecting and analyzing papers with contributions to a constructivist approach of science-policy interfaces in the WEF nexus literature. In general, the framework adapts four pillars suggested by Jasanoff to nexus-related issues; namely, framing, vulnerability, distribution, and learning. These pillars provide a broader look at both science and policy formulation. This does not suggest that one must follow all the pillars to the letter in terms of practical implications, rather that it is crucial to be aware that the nexus is much more a point of view than a replicable truth [14].

**Figure 1.** Conceptual framework of the 'nexus of humility' as a pathway to Sustainability. Adapted from Jasanoff [15,16], Leach et al. [8,17], and Urbinatti et al. [14].

Humility is the key to dealing with layers of ignorance in decision-making, considering both the limits of scientific knowledge and the limits of science to guide policymaking [16]. This general idea is complementary to the existing emphasis on 'sound science' as a basis for policymaking on the 'WEF nexus' [14].

To address the conceptual aspects of 'hybridity' and 'humility' in governance processes holistically (i.e., politically and institutionally), we opted to include in the framework some aspects of governance suggested by the PTS approach [17]. This gives rise to Sustainability, which is understood as the ability to maintain stability, durability, resilience, and robustness [8,17]. However, it is necessary to understand what Sustainability means in a given context, and this task should be undertaken as part of governance [17]. It is precisely the relationship between local and global that is implicit in the concept of Sustainability, which associates it with the nexus debate. Resources such as water, energy, food—and health—cannot be seen only on a global scale; they require the context and the people involved in it to reveal the complexities at stake.

Nevertheless, bargaining on PTS is necessarily a political process. It therefore requires knowledge beyond the ability to conduct scientific analyses, namely knowledge of the diverse and deliberately inclusive kind. Overall, the awareness of power relations in the production of knowledge and in the decision-making process encourages humility in governance [18]. Based on Leach et al. ([18], p. 63), three questions can guide the analysis of PTS:

Who are the actors and networks articulating the narrative? How is incomplete knowledge dealt with? Which dynamic properties of Sustainability are prioritized? Answering these questions, according to the authors, requires reflexive processes in order to make explicit that any assessment is partial and positioned depending on the social, economic, and political subjectivities of the analyst. This paper seeks to use and adapt this theoretical reference suggested by the PTS approach for the analysis of governance processes in the Brazilian context.

### 3. Methodology

This work involves case study research and inductive–deductive narrative inquiry. We do not delineate a specific mode of narrative analysis; instead, we seek to introduce 'holistic' and 'categorical' dimensions as well as 'content' aspects [19]. Our version of 'narratives' in fact involves stories formed from particular frames of a given system that are determined by actors, networks, and institutions that define problems and promote pathways to their solutions [18]. The narratives, therefore, justify specific types of actions, strategies, and interventions, some of which are supported by governance processes that shape pathways of interactions among social, technological, and environmental systems.

For primary data, 20 key stakeholders were interviewed using open-ended and semi-structured formats. These interviews were conducted both individually and in group discussions, focusing on the Brazilian Health System (SUS in Portuguese). Group discussions had more open formats while the individual ones were semi-structured. The content of the interviews was transcribed, and the qualitative data were analyzed using codes related to the themes of the nexus of humility framework and the relationship with PTS and WEF nexus concepts. Among the interviewees were municipal secretaries, City Hall technicians, coordinators of Basic Health Units (BHU), community health workers (CHW), environmental promotion workers (EPW), and health service users. Additionally, participants were observed at the study sites to formally record contextual information. As secondary data, other studies on the subject as well as available governmental information such as official documents and website pages (Instituto Brasileiro de Estudos e Apoios Comunitários, 2020. Programa Ambientes Verdes e Saudáveis. Available in: http://www.ibeac.org.br/1339-2/ (accessed on 28 July 2020)) (Prefeitura de Guarulhos, 2017. Documento norteador para a Atenção Básica do Município de Guarulhos. Available online: https://www.guarulhos.sp.gov.br/sites/default/files/DOCUMENTO%20NORTEADOR%20PARA%20ATEN%C3%87%C3%83O%20B%C3%81SICA%20DO%20MUNIC%C3%8DPIO%20DE%20GUARULHOS_20_12_17.pdf (accessed on 1 August 2020)), and material from media (Guaruhos Hoje, 2017. Saúde lança programa para melhorar condições socioambientais da cidade. Available online: https://www.guarulhoshoje.com.br/2017/08/05/saude-lanca-programa-para-melhorar-condicoes-socioambientais-da-cidade/ (accessed on 15 May 2020).) (Rede Brasil Atual, 2010. O Jardim Helena é um lugar esquecido. Available online: https://www.redebrasilatual.com.br/cidades/jardim-helena-e-um-bairro-esquecido-dizem-moradores-depois-de-60-dias-com-casas-alagadas/ (accessed on 23 July 2020)) were reviewed. The data analysis was based on qualitative methods, as described by the following five steps [20,21]: (i) organization and preparation of the data, (ii) obtaining a general sense of the information, (iii) implementing a coding process using Atlas.ti software, (iv) division into categories or themes (initially 34 codes, and finally 6 main categories of narratives), and (v) interpretation of the data through the lens of the 'nexus of humility framework' as the mode of governance under PTS.

The cities of São Paulo and Guarulhos, located in the Metropolitan Region of São Paulo, were the case studies for this research. In São Paulo, we analyzed the GHEP with in-depth analysis of the Jardim Maia BHU located in the east zone of the city. In Guarulhos, we analyzed the EHP by focusing on the Novo Recreio BHU, which is located in the northwest zone of the city. These locations were selected as the cases of interest in this work for different reasons. The selection of Guarulhos was a direct consequence of the project 'Resilience and Vulnerability at the Urban Nexus of Food, Water, Energy and the Environment (ResNexus) (Project funded by the São Paulo Research Foundation

(FAPESP), grant number: 2015/50132-6. Duration: 1 March 2016–28 February 2019)', from which this research is derived. A previous study at the same BHU [10] significantly contributed to this study. The selection of São Paulo is attributable to an interview with the program coordinator of the Jardim Maia BHU, the outcome of which pointed to its value in analyzing how vulnerabilities in relation to water, energy, and food can arise. We approach the two case studies not as a direct comparison, but to understand similarities and differences. Figure 2 shows the two studied regions.

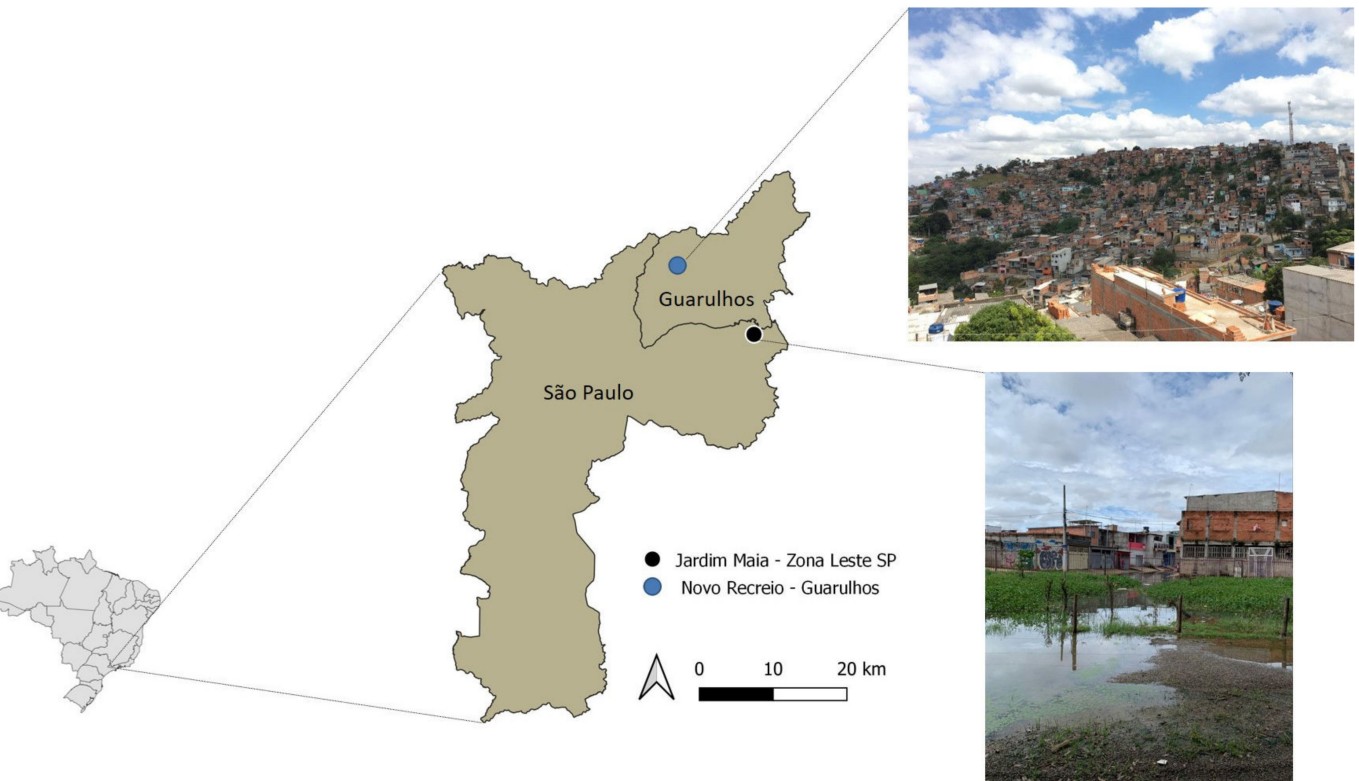

**Figure 2.** Map of the study areas. Source: Authors' elaboration.

### 3.1. The Green and Healthy Environment Program in São Paulo

The GHEP was inaugurated in 2005 in the municipality of São Paulo in response to the demand for public policies that could holistically consider environmental issues, health promotion, and quality of life, especially regarding the marginalized populations living on the outskirts of the municipality. To this end, an alliance was signed between the Municipal Secretariat of Health, the Municipal Secretariat of Green and Environment, and the Municipal Secretariat of Assistance and Social Development in conjunction with the United Nations Programme for the Environment. The objectives were to train workers to focus on socio-environmental issues while acting in the interests of the population, generate spaces for co-management to allow the community to face environmental health risks, and develop an agenda of integrated health and environmental actions [22]. In 2008, the Municipal Secretariat of Health incorporated the GHEP as a program within the so-called Family Health Strategy, a leading SUS program. The GHEP was linked to the Coordination of Primary Care with the task to contribute to the implementation of integrated public policies in the municipality and foster empowerment and community participation [23].

### 3.2. The Environment and Health Program in Guarulhos

The EHP was announced in 2017 with the objective of building participatory and collaborative processes for sustainable development through integrated agendas conducted by secretariats and sectors of the local government, involving the Municipal Secretariat

of Health, Municipal Secretariat of Education, Municipal Secretariat of Environment, Municipal Secretariat of Public Services, Zero Waste Program, local health councils, among others. Based on the Sustainable Development Goals, the program focuses on interventions in territories and the empowerment of communities, helping to elaborate public policies aimed at creating a 'healthy' and 'sustainable' city. Although it was launched via an official event at the City Hall of Guarulhos, data regarding the program are unavailable. The data used in this study were thus sourced directly from the Municipal Secretariat of Health, particularly the Department of Integrated Health Practices, CHW and health service users.

## 4. Results

In this first part of the results, we will show the analysis of the primary data collected during the study. The thematic analysis showed the main codes that indicated the narratives involved in the two municipalities. The 34 codes (in parentheses the number of mentions of each code) defined through the thematic analysis were: references to GHEP (4); references to EHP (4); Sustainable Development Goals (1); sewage treatment (1); SUS values and principles (2); construction of indicators for the programs (4); water, energy, and food (1); access to transport (5); basic health care (2); participation mechanisms (6); expectation for participation (6); risk area (5); vulnerability (1); selective waste collection (1); recycling (1); financial focus on health (4); waste (1); waste management in health services (1); power relations (4); distrust among the public and in participation (6); distrust in government (6); administrative focus (4); barriers to action development (4); flooding (5); focus on development of the neighborhood (5); focus on diseases (2); disease prevention (2); access to water (1); gardens and food (1); intersectorality (3); knowledge production and educational aspects (6); socio-environmental focus on health (1); reference to territories (5); and outputs and assessments (4).

The codes were grouped into the following six main narratives according to the themes relevant to this study:

(1) Environmental and social determinants of health;
(2) Health prevention and promotion;
(3) Intersectorality;
(4) Politics and economy;
(5) Territory;
(6) Learning and participation.

Figure 3 shows the frequencies of these narratives in the transcribed documents.

The results show that 'environmental and social determinants of health' narratives were more prevalent in the discourses of the interviewed actors (26%), followed by narratives on 'learning and participation' (20%), 'politics and economy' (18%), 'territory' (17%), 'health prevention and promotion' (10%), and 'intersectorality' (9%). These narrative groups expanded our understanding of the issues related to water, energy, food, and the environment at the case study locations. In other words, considering the material in a way organized by central narratives is the first step in deepening the discussion within the framework proposed here. Table 1 is based on both primary and secondary data analysis. This shows the correlations presenting dimensions of framing, vulnerability, distribution, and learning.

In addition, aspects suggested by the PTS approach, such as political entities and spaces, structures and practices, power and knowledge, uncertainties, and history, politics and context, were covered.

Notably, the flexible nature of the concept of PTS, as well as the view we propose to the WEF nexus, better explain how the main actors are involved and the priorities in the debate on Sustainability. Nevertheless, the incomplete knowledge of policymakers often encounters community practices that help to direct knowledge production. We highlighted narratives that include elements that reinforce the nexus point of view in the vulnerable contexts of the two municipalities. These are the results of our analysis of the secondary

data collected for the two municipal programs. Figure 4 shows the different axes of the two programs' official documents closely related to the nexus.

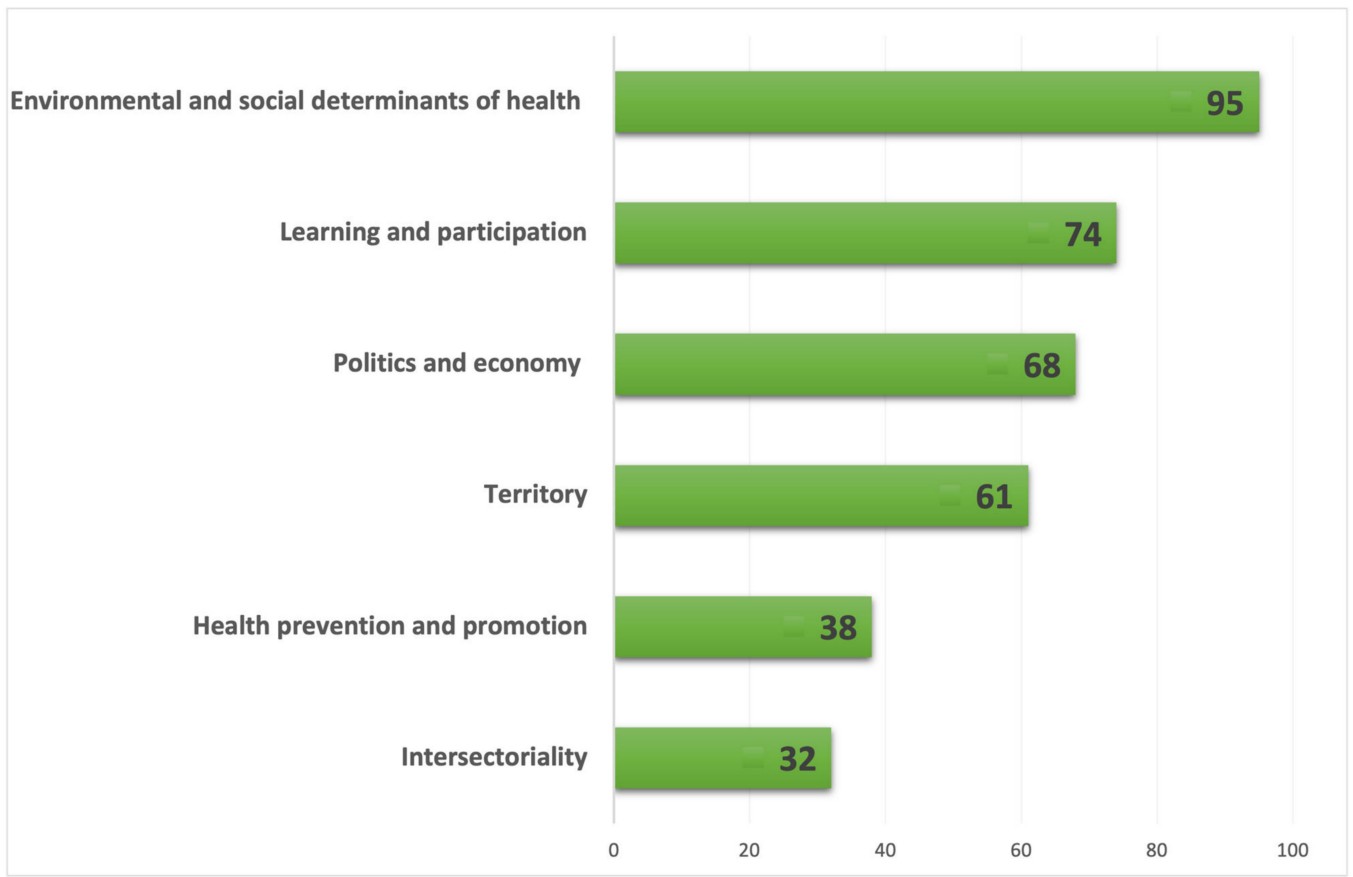

**Figure 3.** Frequency of narratives. Source: Authors' elaboration.

**Table 1.** Nexus of humility between WEF as a pathway to Sustainability from the case studies.

| | São Paulo (GHEP) | Guarulhos (EHP) |
|---|---|---|
| FRAMING Correlations with intersectorality narratives and politics and economy narratives * | -(What) water, energy, food, health, and the environment in a peripheral context (Jardim Maia and region); (How) as thematic axes of a municipal health program; (By whom and for whom) promoted by the Municipality of São Paulo and partners aimed at the local population<br>-Entities: Municipal Secretariat of Health, Municipal Secretariat of Green and Environment, Municipal Secretariat of Social Development, Health Family Association, Santa Marcelina Institution, Santa Catarina Association, Social Responsibility Institute of Albert Einstein Hospital, Monte Azul Association, Federal University of São Paulo, and Paulista Society of Medicine Development<br>-Spaces: Coverage throughout the municipality's health network, particularly in five Regional Health Coordination zones: north, west, central-west, south, and east<br>-Structures: Program within the Department of Basic Health Care; 963 items of healthcare equipment available with the municipality; FHP operates in at least 269 BHUs; 237 Environmental Promotion Workers<br>-Practices: Program developed between 2005 and 2008, during which 5000 CHW with environmental themes capacitated; as of 2018, 205 projects in progress and 473,368 individuals involved during the year; GHEP at an advanced stage of policy evaluation and possible corrective action implementation | -(What) water, energy, food, health, and the environment in a peripheral context (Novo Recreio); (How) as thematic axes of a municipal health program; (By whom) promoted by the Municipality of Guarulhos and partners aimed at the local population<br>-Entities: Municipal Secretariat of Health (departments, units, and services), Municipal Secretariat of Environment, Municipal Secretariat of Public Services, Municipal Secretariat of Education, SUS School, Municipal Councils, and Health Unit Management Councils.<br>-Potential partner entities: University of Guarulhos, School of Public Health/USP, Faculty of Medicine/UNINOVE, NGO Ecoficina, NGO Ecosocial Água Azul, Cooperatives of Collectors, and private initiatives<br>-Spaces: Municipality divided into four health regions: Central, Cantareira, São João/Bonsucesso, and Pimentas/Cumbica; EHP not implemented in an expanded form as a program, but intersects with other program (such as Integrative and Complementary Health Practices) popular in much of the city's territory (including Novo Recreio)<br>-Structures: Department of Integral Health Care responsible for implementing health policies in territories, setting standards, and providing institutional support in a centralized manner; 69 BHUs in the city, 48 of which fit the Basic Health Care guidelines; BHU councils comprise community members, including Novo Recreio<br>-Practices: Program announced in 2017; capacitated many technicians and CHW; currently devising the mission and indicators; official flagging off scheduled for the year 2020; EHP in the transition phase between 'agenda setting' and program and decision-making; seven working meetings held with groups involved in the program; in the next stage the Department of Integral Health Care will invite members involved in the GHEP in São Paulo to share experiences |

**Table 1.** *Cont.*

| | São Paulo (GHEP) | Guarulhos (EHP) |
|---|---|---|
| **VULNERABILITY** Correlations with environmental and social determinants of health narratives and heath prevention and promotion | -History of the region: Region was occupied for a long time; situated in an area that was formerly home to colonial farms; became urbanized with the arrival of a chemical industry in the 20th century. -History of the program: GHEP was inspired by the values and principles of the SUS created in 1988; GHEP related to the FHP; articulated in partnership with the United Nations Environment Programme. -Environment: Borders a municipality in the East; located in a region near the (polluted) Tietê River -Water: Flooding occurs throughout the year -Energy: Illegally accessed by most residents; difficulties in accessing public transportation during flooding periods -Food: Lack of knowledge about healthy food | -History of the region: Peri-urban area that grew in an unplanned manner during the 1990s; belongs to Cabuçu, a region formerly containing farms that were part of the municipality of São Paulo until the 1920s; Cabuçu well-known as the place of construction of the first concrete dam in the country (1908). -History of the program: EHP was inspired by the values and principles of the SUS created in 1988; program based on the Sustainable Development Goals; previously, four directories available for each of the four health regions mentioned above -Environment: Located in the northwest region, bordering the Atlantic Forest area and the Cantareira State Park -Water: Lack of access and supply -Energy: Illegally accessed by most residents; problems accessing public transport -Food: Lack of access to fresh and healthy food. |
| **DISTRIBUTION** Correlations with territory narratives | -Resident population in areas near the Tietê River are the most negatively affected -Problems solved with the assistance of the Jardim Maia BHU in conjunction the sub-City Hall through the Family Health Program (FHP) and GHEP, together with the Santa Marcelina Hospital -Power: Central hierarchical structure, starting from the Department of Basic Health Care; capillary network in the territories; efforts for collective production of knowledge among technicians, coordinators, Environmental Promotion Workers, and the community; experience in territories important for the co-production of knowledge, and to encourage adaptive and reflexive learning processes | -The entire population reports difficulties entering and leaving the neighborhood, especially on rainy days, when public transport also stops running -Important institutions regarding solving resource-related issues are the Novo Recreio BHU (through the FHP), the Nazira Abbud Zanardi Municipal School, and the Mothers Club Association -Power: Power still fairly centralized in the Department of Integral Health Care; little progress so far due to more urgent demands; process of capacitating CHW based on reflexive assumptions of knowledge production; principle of 'nobody makes anybody conscious' applicable but necessary to collectively build consciousness for sustainability; council meetings used for agenda-building at the territory |
| **LEARNING** Correlations with learning and participation narratives | -Constant use of sodium hypochlorite to purify drinking water and water used for household chores -Reuse of leftover food from street markets that take place twice a week -Development of food storage techniques during the flooding season (e.g., storing food at a height) -Donation and sharing of basic-needs grocery packages -Environmental Promotion Worker actively assists the BHU and is one of the main channels of communication with the community; household visits conducted; the Discussion Group 'Harvesting Fruits' is popular -Uncertainties: recognized and related, for example, to arboviruses and flooded areas; mapping system proposed by the municipality used as a technical approach to solve vulnerabilities and enhance territory's potential | -Local food production -Household water storage and rational use of water -Solid waste recycling -Alternatives to public lighting -Practices to overcome neighborhoods inequalities and underdevelopment regarding water and energy connections and sharing resources between households -Acting manager of the BHU Novo Recreio and one of the CHW involved in the capacity building process proposed by the EHP in 2017 -Uncertainties: Contingencies needed in the political process due to uncertainties in governance; arboviruses and vulnerabilities of territories also identified as uncertainties |

Source: The authors, based on both empirical data and Melo [24] and Giatti et al. [10]. * The narratives may be correlated with other quadrants of the table; it is just a way to indicate the strongest connections.

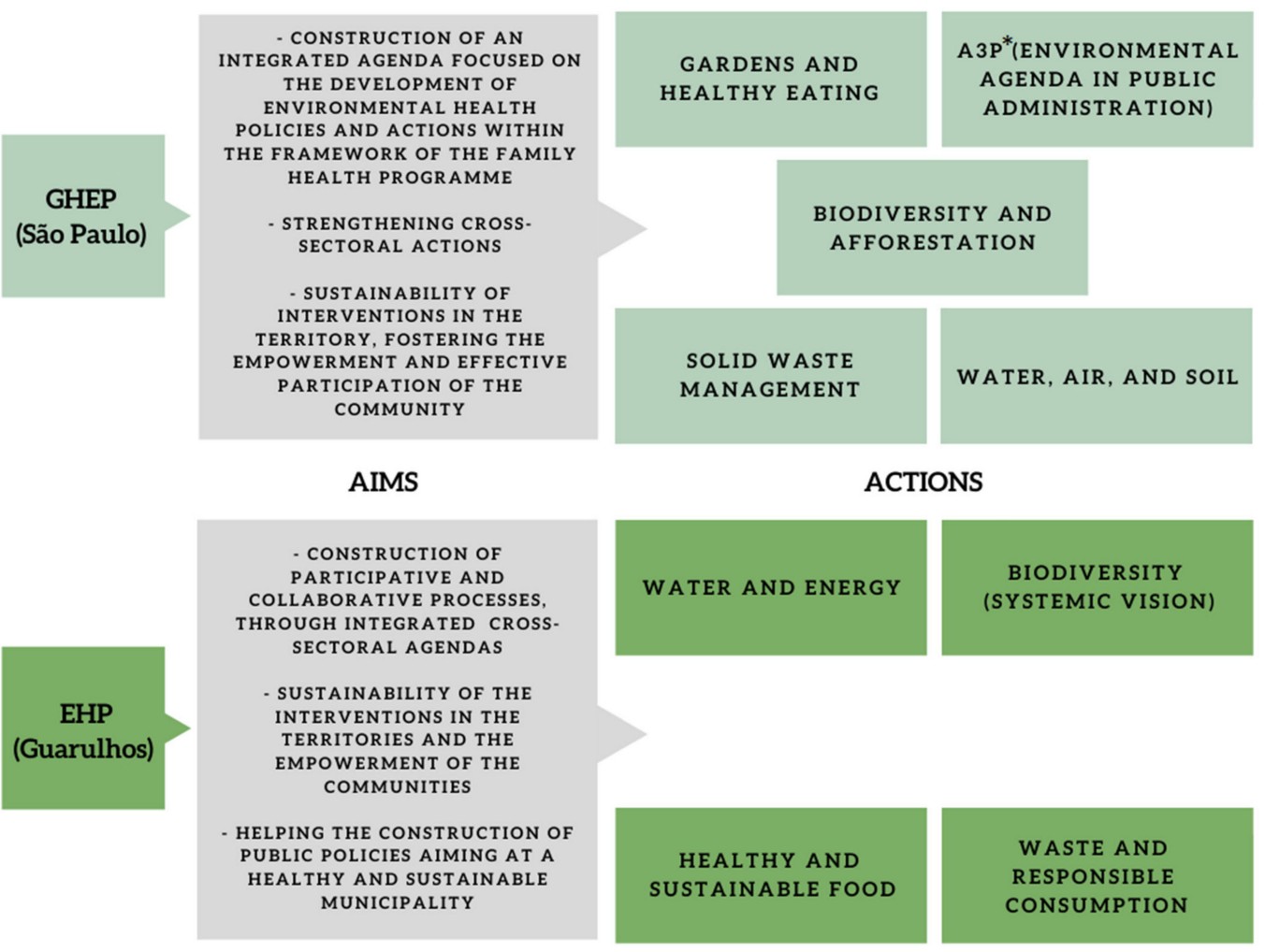

**Figure 4.** Axes are closely linked to the nexus approach in the municipal programs. *A3P is a program from the Brazilian Ministry of Environment that stimulates public institutions of different levels to implement sustainability practices. Source: Authors' elaboration based on municipal documents (Campos, R. C. C. M. 2017. Ambienta Saúde e Objetivos Globais para o Desenvolvimento Sustentável. Visual presentation during the launch event of Ambienta Saúde, Centro Educacional Adamastor, Guarulhos.) (Prefeitura de São Paulo, 2010. Programa Ambientes Verdes e Saudáveis—PAVS. Secreatria Municipal de Saúde de São Paulo. Available in: https://docs.bvsalud.org/biblioref/sms-sp/2015/sms-10219/sms-10219-6878.pdf (accessed on 12 April 2020)).

## 5. Analysis of the Narratives and Discussion

The six groups of narratives interpreted on the basis of the defined codes corroborate with elements from the theoretical backgrounds of the 'nexus of humility' approach. First, the narratives of the interviewees helped us to frame opportunities to discuss the 'nexus dimension' in both municipalities.

Second, the scope of the PTS allows framing the narratives of a given system regarding its environment and delimitations; the structures and functions are essential. Thus, framings and multiple narratives can co-exist while being implemented by different actors and co-produced with institutions based on power and distributed knowledge. Governance itself shapes the prevailing frames and the manner in which they are negotiated, elevating the properties of a system on a temporal scale [17]. This is how possible innovative pathways to local Sustainability are derived from governance processes that emerge on the horizon of practices and policies.

To better interpret Sustainability in these two municipalities, it is necessary to understand the context in which these actions take place. São Paulo is one of the main economic, financial, and political centers of Latin America [25]. The growing urbanization of the peripheries in the last century was very rapid, and approximately 15% of its population (1.5 million people) live in precarious conditions today, most of them occupying areas illegally [26]. According to the Atlas of Social Care [23], 24,634 and 5816 households in Itaim Paulista are classified as highly vulnerable and extremely vulnerable, respectively. The same source lists the region of São Miguel Paulista as having 13,509 highly vulnerable and 12,892 extremely vulnerable households. Being a small area within Jardim Helena and between Itaim Paulista and São Miguel Paulista, Jardim Maia does not appear in the list. In any case, these data characterize all these areas.

Guarulhos, located in the São Paulo Metropolitan Region, is the second-largest city in São Paulo state in terms of population, home to more than 1.4 million inhabitants according to IBGE Cidades [27]. The increase in the urban population in a relatively short time has brought about numerous challenges for the subsequent municipal governments. Some examples include deep social inequalities and environmental problems, such as the small-scale sanitation infrastructure, and the serious risks associated with pollution and poor water supply to the population. The Novo Recreio neighborhood is considered part of the Cabuçu region, and suffers from exactly these problems due to the lack of investments through planning and policies.

Given this scenario, the social vulnerabilities became evident from the interviewees' speeches (The transcribed passages presented here were translated to avoid altering the meanings of the interviewees' narratives. However, we recognize that there is a risk of this happening in any translation exercise.). For example, one of the CHWs and resident of the Novo Recreio neighborhoods, identified here as I1, said the following:

I1: *'We thought—I speak for myself—we thought that with time it would settle down here, I would pay for my house, everything would stay beautiful! But that's not how it happened!'*

The narrative of underdevelopment was noted at many times in the conversations as a thread running through many of the problems related to quality of life, including access to resources, as listed in Table 1. An interview with another CHW (identified as I2), also a resident of the neighborhood, exposes the environmental vulnerability.

I2: *'I'm glad you came now. If you'd come a week ago, you couldn't have come here. A landslide happened up there. That little stream over there flooded and destroyed the area. A tree fell, and we had to travel on foot'.*

In the case of São Paulo, we witnessed problems of accessing the flooded area in the region of Vila Seabra during this study. This area is under the care of the Jardim Maia BHU. It was not possible to enter the area. One of the residents of the region (I3) narrated how this problem, rather than the perceived lack of resources, affects them.

I3: *'We have enough energy for our use. Transport is another matter, however; we find it difficult to leave home (many of us have to face flood waters without any protection)! When floods occur, many families end up losing food if it is not stored properly and due to lack of planning. Some people face financial hardships and run out of food, and they can only count on neighbors to help!'*

The theme of flooding is recurrent in our interviews with the São Paulo residents. Another interviewee from the BHU (I4) said the following:

I4: *'[ . . . ]when the river floods, it brings back sewage with it, increasing the risk of diarrhea and vomiting. If you pass through the area, you can smell the sewage'.*

Notably, a relationship of interdependence exists between the environmental issues, vulnerability, and health determinants of a territory, thereby generating a characteristic 'nexus perspective' for each region. It is in this sense that the narratives collected in this study allow us to understand the importance of the BHUs as the protagonists of

the learning process concerning urban Sustainability. These narratives indicate resilient practices occurring in the communities. In other words, this is exactly where the limits of both policy-makers' and scientists' expertise and community knowledge lie. Thus, the two municipal programs assume vital dimensions regarding the institutional paths relevant to PTS. In the case of São Paulo, as listed in Table 1, an interview with an important civil servant from the Municipal Secretariat of Health (I5) revealed that despite the hierarchy laid down in the guidelines and laws of the Secretariat, the territories create their own community-based action networks.

> I5: *'You want to know how everything down there [in the communities] is going? I don't know and I don't want to know! All I know is that they happen. Things take their course, their direction. There are professionals who manage this; now, it doesn't have to be from top to bottom. There's no such thing, there's no such thing! There are rules, there are guidelines, but the territory demands its own pathways'.*

The health policy underwent some of the most radical changes in the decentralizing process during re-democratization since 1988 with the new constitution, as the government guaranteed the creation of a universal and decentralized public policy system, the SUS, with the responsibility of service provision being assigned to states and municipalities [28].

The narratives related to nexus can be identified in both policies, as seen in Figure 4. Most of the projects implemented by the GHEP teams included community gardens. Many, especially the elderly, displayed the willingness to tend to the land and produce their own food. Community gardens that originate from public programs, such as GHEP, have a high chance of sustaining themselves. It is evident that such gardens are only possible due to the efforts of the group members and community participation; however, the involvement of public agencies and other partners, through donations of seedlings, tools, space, and technical collaboration, increases the probability of long-term success. GHEP relates to other programs such as Traditional Medicine, Homeopathy and Integrative Health Practices, Health of the Indigenous Population, Integral Health Care for Victims of Violence, Health for Children and Adolescents, Adult Health (for those with hypertension and diabetes), Health of the Elderly, Mental Health, Nutrition, and Family Health Support Centre, among others.

Consider, for instance, the community garden created at the Jardim Maia BHU in 2018 for learning purposes. It was located in the municipal school next door. It was eventually terminated due to a lack of commitment. Currently, the main activities of the BHU are related to household visits encouraging responsible consumption practices, personal hygiene, disease prevention, and providing incentives for the correct disposal of batteries, cooking oil, and medical waste among others. Discussion groups called 'Harvesting Fruit' are held on Tuesday of every week, which focus on topics related to water use and quality, food quality, and sustainable practices in the region. The BHU team is led by the Environmental Promotion Worker who participated in the 'Earth Hour' competition organized by the Santa Marcelina Association and inspired by the non-governmental organization WWF in 2019. The elderly users' group at the BHU won third place, which cemented their interest in issues pertaining to climate change and Sustainability (Available online: http://www.aps.santamarcelina.org/iniciativas-ambientais-sao-premiadas-na-hora-do-planeta/, accessed on 15 May 2020).

Household visits are also part of the regular practice in Guarulhos. However, no specific posts for EPW exist at the Novo Recreio BHU, and the CHWs are responsible for this function. The aims of the EHP intersect with those of Integrative and Complementary Health Practices and have resulted in many new gardens at the BHU to be used by the local population. The BHU at Novo Recreio did not have any gardens before the initiation of our research. Many community gardens were later created as the Novo Recreio BHU realized the therapeutic and health alternatives that could be availed by involving patients and residents as well as the local school, EPG Nazira Abbud Zanardi, which planned to use the garden as a pedagogical tool for its students and a therapeutic space for its employees. The community garden hosted by the school was also the site of a master research project and

the international ResNexus Project, which was supported by EHP. Although the volume of the garden's produce was too small to benefit the entire neighborhood, it worked as a didactic laboratory that provided space for meetings and food for thought regarding the development and importance of a community garden.

As previously mentioned, the community garden not only served for research purposes, but also for validating popular knowledge, promoting community empowerment, and fostering local partnerships in the neighborhood and with public agencies. In addition, through the lens of the ResNexus Project, the community garden represented, an example of synergy between the water, energy, and food sectors; referring to Hoff [12], local food production reduces waste generation, since production circulates within the neighborhood, thereby eliminating the transportation required for conventional distribution and energy for storage, thereby reducing losses. However, certain conflicts were observed regarding the continuity of the activities. According to one interviewee (I6), the garden prospered when the research group was involved, but once the research ended, the group failed to work collaboratively.

> I6: *'So, what did we get out of it? While it was with you, the access was different, right? When it was just us from the community, the interest seemed to . . . it was different, we felt . . . '.*

In addition, some conflicting relationships arose in relation to change in the management of the BHU, as some were more predisposed to solving environmental issues, while others were less so. These activities are related, either directly or indirectly, to broader governance processes. To understand governance nuances, we sought to relate aspects of 'policy cycle' previously mentioned. According to Frey [29], to understand 'policy analysis'—including 'policy cycle'—in the Brazilian context, it is necessary to consider the unstable dynamics of the country's political institutions and processes. These dynamics are evident in the case of EHP, for instance, since this program did not develop exactly as announced by the government at the time. These aspects affected the survey data, as no official publications exist on the program to date. Although we have not responded in detail to each of the policy analysis categories identified here, the data identify the aspects of each at the macro level.

Table 1 illustrates the similarities between the two policies as well as their crucial differences. An important similarity, on the one hand, is that the EHP is inspired by the GHEP model. This can contribute to developing a metropolitan dynamic of sharing experiences on Sustainability in public health. On the other hand, some remarkable differences exist in relation to political–administrative practices and the prevalent institutional structures, revealing the 'policy cycle' stages distinctly. That is, while the EHP is at the program design stage, seeking to implement the policy, the GHEP is at the advanced stage of policy evaluation, correcting eventual shortcomings. These different temporal contexts imply different network configurations: while the GHEP is part of regional, national, and international collaborative networks, the EHP has only begun to establish basic local partnerships.

In the case of the GHEP, despite the top-down hierarchical traits of the program guidelines, the activities in the capillary territories have harnessed power relations, conflicts, and knowledge production. This contrasts with the predominance of narratives related to learning and participation, indicating that there are some fundamental lessons to be learned from local social practices of vulnerable communities regarding how to deal with interdependent scarcities [30]. In the EHP, the activities have not yet formally expanded into the territories, given that they are concentrated under the aegis of the Municipal Secretariat of Health. The demands around medical waste have become an urgent issue for the current government and have changed the course of the process. However, it must be recognized that many sustainable practices are already underway in the territories, including in the Novo Recreio, as CHW together with the BHU Council and community members have made significant strides in the search for improvements to the region. Therefore, we conclude that there is a degree of humility present in the two programs, insofar as to create knowledge under diverse perspectives.

## 6. Concluding Remarks

By proposing this analysis of the implementation of two sustainability-focused health programs based on the framework we call the 'nexus of humility' and combining it with the PTS governance approach, we sought to move closer to a humbler appreciation of the advances and barriers regarding Sustainability governance in urban contexts. While these two programs are not necessarily based on the nexus approach, we could recognize synergies and opportunities for 'nexus thinking' in the municipal guidelines and respective implemented actions, based on both primary and secondary data. In particular, the discussion of 'the nexus' proved highly relevant concerning the connections between the democratic and reflexive components of the programs. Public health is an important supporting pillar for the development of synergetic relations of Sustainability in urban areas. Moreover, the examples of the investigated municipal health programs, which are grounded in the values of the Brazilian Health System (SUS), provide opportunities to think about the water–energy–food nexus from a public health perspective and its importance for simultaneously making vulnerable communities more resilient and healthier. In addition, the nexus thinking through the data analyzed in this study contributes to the understanding of pertinent reflexive connections between urban vulnerable communities' dynamics and the global problem of the intertwined scarcity of water, energy, and food.

PTS governance is an effective analytical tool to identify narratives related to 'the nexus' in the two municipal programs. First, it enabled us to better understand processes on the community level not always explicit in Brazilian urban governance. Second, we understood 'the nexus' as a holistic point of view expressed in the main six narratives identified during the conducted interviews. Thus, the main issues encompassed by the nexus approach, such as access to resources and quality of life, can be correlated with the debate on environmental and social determinants of health. The predominance of these narratives indicates that the nexus approach can become more robust if it includes the public health focus. Moreover, although the programs are based on participatory approaches, obvious traces of hierarchy and control were evident, which does not necessarily diminish the distribution of benefits; however, it does show that government action has its limits. Despite the predominance of narratives related to learning and participation, the very structure of the government has created constraints.

The results of our research will facilitate positive transformation beyond these programs and the territories in question. Yet, such a transitional process demands innovative scientific framing and participatory governance approaches that are able to explore the synergies of urban Sustainability, particularly related to the WEF nexus. Nevertheless, pursuing Sustainability in large urban centers of the Global South, such as São Paulo and Guarulhos, which are characterized by huge social inequalities in relation to access to resources, will always be challenging. This study sought to contribute knowledge using assumptions that reflexively 'open up' concepts and theories used in recent years. Moreover, we suggested possible ways to rearrange them in Brazil's unique context, apart from contributing to innovative developmental perspectives for cities in the Global South.

**Author Contributions:** Conceptualization and methodology, A.M.U. and L.L.G.; investigation and data curation, A.M.U., S.L.O.-H. and C.M.d.C.; validation and formal analysis, A.M.U.; writing the draft version, A.M.U. and L.L.G.; review and editing the final version, A.M.U., S.L.O.-H., C.M.d.C., K.F., P.R.J. and L.L.G.; supervision K.F., P.R.J. and L.L.G. All authors have read and agreed to the published version of the manuscript.

**Funding:** This research was funded by FAPESP—the São Paulo Research Foundation, process numbers 2016/25375-5 and 2018/12155-2.

**Institutional Review Board Statement:** The study was conducted in accordance with the Declaration of Helsinki and approved by the Ethics Committee for studies involving humans of the School of Public Health, University of São Paulo in 2017 (decision number 3.733.306), and amendment in 2019 (decision number 3.792.122).

**Informed Consent Statement:** Informed consent was obtained from all subjects involved in the study.

**Data Availability Statement:** The data presented in this study are available on request from the corresponding author.

**Acknowledgments:** The authors acknowledge the support provided by FAPESP (process numbers 2015/03804-9, 2016/25375-5, and 2018/12155-2). LLG also acknowledges CNPq—the National Council for Scientific and Technological Development (process number 314947/2021-3).

**Conflicts of Interest:** The authors declare no conflict of interest.

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
