# Peer review of "‘Nexus’ Narratives in Urban Vulnerable Places: Pathways to Sustainability via Municipal Health Programs in Brazil"

_world, doi:10.3390/world4010002_

Round 1

Reviewer 1 Report

Related Work:

It has a good collection of the literature survey, but how is not explained in the methodology part.

Method:

It has a good collection of methods. But the inputs and outs from the method applied are not clear in the manuscript. Whether it is primary data or secondary data. The methods adopted for collecting the data, as well sampling techniques may be elaborated.

Results and Discussion:

The number of respondents are only twenty, need increment!

Author Response

Dear reviewer,

Thank you very much for your appointments. We made use of your review to improve our paper.

Concerning the basis of related work, we gave some details of the literature survey used to propose the ´nexus of humility´ framework. We have improved the description of all methods and approaches applied in the research. For that matter, we gave more details and explanations about what and how secondary data was used, and we also made inputs for better explaining the acquisition, organization, and analysis of primary data. These improvements were mainly inserted in the methodology section. Regarding the appointment on the number of respondents, we justified our sampling, besides this, we argue that the new inputs on the interplay of secondary and primary data makes it clearer to understand that our sample is reasonable.

Reviewer 2 Report

 The paper focus on the urban vulnerability and public health in two cities in Brarzil. This focus of the paper could contribute well journal World ongoing discussions. Although the current version of the paper require additional work.

Theoretically this paper has ambition to elaborate ´humility nexus´ towards sustainability pathways (Figure 1 ). This framework needs more clear explanation in bringing together the existing approaches (e.g. Jasanoff´s technologies of humility, and Urbinatti et al. water-energy-food nexus).  Currently it remains unclear what is the wider potential in looking sustainability transition as humility, and the use of ´nexus´ and nexus-thinking should be more clearly articulated in the study.          

The paper is based on the qualitative methods and narrative approach in analysing health programmes in two cities, Sao Paulo and Guarulhos. There are generated 6 themes in analysing the narratives related to health programs.  The structural framework (Tabel 1) of analysing empirical materials puts institutions in forefront and lacks of community-level knowledge, actions etc. in pathways to sustainability. I would argue that the research questions and this framework of analysis do not match well currently for answering all the posed questions, e.g. issues of temporal dimensions and uncertain knowledge.  This requires additional work.

The link to interview materials is good (sub-chapter 5). It requires some explanation that how sub-chapter 4 (results) and 5 are analytically related. The reflection on the current structure in analysing and interpreting empirical material is also necessary. The posed research questions (p. 4-5) should be more directly addressed and answered in analytical discussion and conclusion.   

Author Response

Thank you very much for your contribution in reviewing our manuscript, which gave us good insights into improving the quality.

Regarding the ´nexus of humility´ background, we gave more details and explanations on the applicability of this in our manuscript. For that issue, we also gave some details about how this conceptual basis was elaborated by a bibliographic review in previous research. These compliments are in the ´conceptual framework´ section, mainly on page 3 after figure 1.

To make a better correspondence among the components of the manuscript (structural framework, qualitative data, and discussions), we have adjusted the research questions and objectives, improved details on methodology (also with a better description of secondary and primary data), and reviewed data presentation, discussion, and conclusions.

Therefore, we consider that the improvements also contribute to better organization and explanations of the results and analytical contributions, thus, making better backgrounds to the presented discussion and conclusion.

Reviewer 3 Report

A study with original contribution to literature in the context of the largest developing country of the global south. 

Author Response

Thank you very much for reviewing our paper. We expect that with this last review we are providing a version with an even more concise contribution to understanding the search for sustainability through intersectoral approaches and public health policies in developing countries.

Round 2

Reviewer 2 Report

Authors have mainly adressed my ealier review comments. The contribution of the theoretical framework in thematic analysis still remains a bit unclear. However, the paper seems to be ready for publishing and further discussions.